# Invasion History and Dispersion Dynamics of the Mediterranean Fruit Fly in the Balkan Peninsula

**DOI:** 10.3390/insects15120975

**Published:** 2024-12-09

**Authors:** Mario Bjeliš, Vasilis G. Rodovitis, Darija Lemic, Pantelis Kaniouras, Pavao Gančević, Nikos T. Papadopoulos

**Affiliations:** 1University Department of Marine Studies, University of Split, Ruđera Boškovića 31, 21000 Split, Croatia; 2Laboratory of Entomology and Agricultural Zoology, Department of Agriculture Crop Production and Rural Environment, University of Thessaly, Fytokou St., 38446 Volos, Greece; rodoviti@uth.gr (V.G.R.); nikopap@uth.gr (N.T.P.); 3Department of Agricultural Zoology, Faculty of Agriculture, University of Zagreb, Svetosimunska 25, 10000 Zagreb, Croatia; dlemic@agr.hr; 4Esri R&D Center, 1050 Wiena, Austria; panteliskaniouras@gmail.com; 5Undergraduate Program “Mediterranean Agriculture”, University of Split, Ruđera Boškovića 31, 21000 Split, Croatia; pgancevic@unist.hr

**Keywords:** *Ceratitis capitata*, invasion history, host plants, pest phenology, elevation, establishment

## Abstract

In this article, we have reviewed and analysed all the available information, including in historical records, on the phenology and infestation rate of fruits by the Mediterranean fruit fly (medfly), *Ceratitis capitata* (Wiedemann 1824; Diptera, Tephritidae), in the Balkan Peninsula, to illustrate and understand the path of invasion and spread dynamics in the northern Mediterranean region and Central Europe. After the medfly was first discovered in an area of the Aegean Sea in 1915, the pest was then detected in the Peloponnese in the early 1930s, along the entire Adriatic coast in the 1950s, and has been found in the Black Sea area since 2005. Since 2000, a significant increase in the frequency of detections has been recorded in the interior of the Balkan Peninsula, including occasional outbreaks in areas with unfavourable climatic conditions for overwintering, which seems to reduce the reproduction of the pest in summer and autumn. In the last 20 years, the medfly has spread to more northern areas and has been detected at higher altitudes. There are 25 host plant species that have been reported as host plants of the medfly in this area. Considering the extremely high invasiveness of the medfly and its wide distribution in several areas of the Balkan Peninsula with different climatic conditions, we can assume that it is adapting to new climatic conditions and infesting new host plants.

## 1. Introduction

The Mediterranean fruit fly (medfly), *Ceratitis capitata* (Wiedemann) (Diptera: Tephritidae), is considered one of the most important invasive phytophagous pests worldwide, infesting more than 300 different plant species [1,2]. Potential host plant species for the medfly include a wide range of genera and families. However, the preferred hosts belong to the families Rosaceae, Rutaceae, and Moraceae [2]. The sequence of maturation in preferred and less-preferred host plants enables the medfly to develop several generations from spring to autumn, causing significant infestation to host plant fruit and economic damage that threatens fruit production and trade. The medfly is distributed on five different continents in temperate, subtropical, and tropical climates [2,3,4,5]. Originally from eastern tropical sub-Saharan Africa [6], after it was first discovered in the Iberian Peninsula in 1842, the medfly spread to all the Mediterranean countries, first to Italy in 1863, followed by France in 1885, and then to other temperate areas along the Mediterranean coast [7,8]. Its significant invasive properties (ability to spread and adapt) and its high reproduction rate are significant for the establishment of medfly populations in new areas [7] and the significance of the medfly’s invasive properties mean that it has an ecological impact [9].

Under optimal climatic conditions, the medfly completes its life cycle in 21 to 30 days [10]. Depending on the temperature, the eggs hatch in 1.5 to 4 days. The larval stage can last six to ten days and the pupal stage, six to thirteen days, if the average temperature is between 24.4 °C and 26.1 °C [11,12]. The lowest development temperature for larvae was reported to be 10.2 °C [13]. Adult activity is reduced or suspended at temperatures around 30 °C [14]. Without behavioural thermoregulation, the lower and upper temperatures that allow for coordinated adult movement are in the range of 5.4–6.6 °C and 42.4–43.0 °C, respectively, but these values vary depending on the age, feeding status [15], and short- and long-term thermal history of the medfly [16,17,18].

The geographical range of the medfly is traditionally considered to extend to 41 degrees north latitude [19,20] and restricted to habitats where temperatures rarely fall below 10 °C [21]. However, breeding medfly populations have been reported in northern Greece [3,22], northern Italy [23,24,25,26,27,28], southern France [29], Austria [30,31], and southern Germany [32]. The detection and monitoring of medfly during the summer in stone fruit orchards in several municipalities in the Friuli Venezia Giulia region did not reveal the presence of medfly [33], assuming that this area represents the northern limit of the pest’s distribution. In addition, several publications report the ability of the pest to overwinter as larvae in infested fruit in cold winter conditions and to survive in temperatures below freezing [3,22]. Recent findings show that the medfly cannot survive outdoors in Vienna (Austria), but can survive in shelters built by humans [34]. The above data suggest that the species is able to complete its life cycle in areas far beyond its traditional range in the north–east. Climate change [35], the phenotypic plasticity of the medfly [36,37], and the ability of the species to recolonise new areas through local transient populations [38], are expected to play an important role in the expansion of the species’ existing global distribution [39]. The Balkan Peninsula is located in south–eastern Europe and includes the southernmost point of Europe (Crete, Greece), extends to Austria and Hungary, and covers a variety of climatic zones, ranging from the Mediterranean coast to the cold continent. To date, there are no comprehensive publications on the invasion history and establishment dates of the medfly in the Balkan Peninsula, which specifically delimits the area along the Adriatic, Ionian, and Aegean coasts and the western coastal zone of the Black Sea. This narrow belt represents a kind of northern distribution area of the established medfly population [40,41], which includes all the historical pest records reported so far.

This study aims to assess the path of medfly invasion along the Balkan Peninsula, from its first introduction into the region to the present day, including its establishment and spread over time, the host plants and their susceptibility to infestation, medfly population levels and phenology in established and newly invaded areas, and predictions in terms of its further spread.

## 2. Source of the Data and Organization of the Review

The search for relevant publications included published scientific articles, abstracts, and short communications, from the Web of Science, Scopus, and Google Scholar databases, as well as NPPO reports, where available, with articles published in English, Italian, German, Croatian, Serbian, Slovenian, Montenegrin, Albanian, and Greek, concerning studies on the geographical area of the Balkan Peninsula in the period 1915–2022.

The analysis of the data is divided into the following topics: geographical and climatic characterisation of the Balkan region, host plants and their importance, historical overview of the invasion and current distribution, pest phenology, and the importance of agroecological conditions for future spread and establishment.

## 3. Analysis of the Data

The section on the geographical and climatic characterisation of the Balkan region contains the most important geographical data, including the prevailing climatic conditions in the Balkan region. The section on the host plants and their importance discusses the role of numerous host plants in the establishment and spread of the medfly. The coastal area of the Balkan Peninsula stands out in terms of the spread of the pest in new fruit intended for commercial and ornamental purposes. The number of available host plants and the ripening sequence of their fruits have a great influence on the biological characteristics of medfly and its impact on fruit production, especially in terms of the importance of commercial and quarantine processes affecting international trade in fresh fruit. From the collected references, we have selected those that specifically deal with a historical overview of the invasion and current distribution of medfly populations. We address this topic based on three historical periods: (1) the first appearance and introduction period, (2) the establishment and spread period, and (3) the further spread within the invaded area. In this section, we discuss the possibility of medfly introduction through infested fruit from host plants, facilitated by human transportation and trade. The surveillance-based paper describes in detail the three phases of the medfly invasion process, namely the introduction, establishment, and spread. The section on the phenology of the pest highlights the seasonal flight time and population size of the medfly in the southern Balkans, where the pest is established, and in newly invaded inland areas of the Balkan Peninsula. The section on the importance of agroecological conditions for future spread and establishment summarises the data collected, analysed, and presented in the previous five sections and discusses the invasion history, agroecological conditions, and future spread of medfly in the Balkan Peninsula and continental Europe. The trend of the medfly invasion is presented in relation to the difference between the elevation and year of capture in different countries (“Graphicon was plotted in the R 4.4. software with the package ggplot2. The trend line was drawn using the ‘geom smooth’ function, with the loessmethod”).

### 3.1. Geographical and Climatic Characterisation of the Balkan Region

The area of the Balkan Peninsula (Figure 1) comprises a number of geographically significant natural areas, including the Adriatic Sea, the Ionian Sea, the Aegean Sea, and the west coast of the Black Sea, which have climatic similarities, as well as the continental area and the mountainous region, which have a considerable influence on the climate of the Balkan Peninsula. The northern and central parts of the Balkans are characterised by cold winters, warm summers, and well-distributed rainfall throughout the seasons. In contrast, the southern and coastal areas are characterised by a Mediterranean climate, with hot, dry summers and mild, relatively rainy winters [42,43,44,45].

Viewed from the north–west, the eastern Adriatic coast is geographically located between Italy in the north–west and Greece in the south–east (39° and 45° north latitude) and includes the coastal areas of Slovenia, Croatia, Bosnia and Herzegovina, Montenegro, and Albania. Starting with 47 km of the Slovenian coast, the largest part of the Adriatic Sea comprises the Croatian coast, which is 1777 km long and, including all the islands, covers around 6000 km of coastline. This is followed by 21 km of the coast of Bosnia and Herzegovina in the bay of the city of Neum, 294 km of the Montenegrin coast, and 476 km of the Albanian coast, which also includes part of the Ionian Sea. This makes a total of 6837 km of coastline and islands. The south–eastern part of the Adriatic Sea borders the upper part of the Greek coastal areas of the Ionian Sea. The Greek archipelago comprises 3000 islands and the coastline are approximately 15,000 km long. The Greek mainland is very mountainous (80% of the area is mountainous). The Black Sea coast spans 225 km in Bulgaria and 300 km in Romania and is bordered by the Danube, which flows into the Black Sea [42,43,44,45].

The area of the eastern Adriatic is predominantly made up of a closed sea, which is connected to the Ionian Sea and via this to the Mediterranean. The eastern coast of the Adriatic, especially the Croatian part, is strongly indented and mountainous along the Slovenian border with Italy to the coast of Albania, with a few exceptions, namely western Istria in Croatia, the hinterland of Zadar County in Croatia, a small part of the Montenegrin coast, and the north of the Albanian coast. The south–eastern part of the Adriatic Sea and the Ionian Sea in Greece includes islands, as well as the mainland and highland areas. The territory of Bulgaria is divided into two parts by the Balkan Mountains. The southern part, especially the coastal part, is characterised by a slightly sandy coastal area, but with extensive forest areas, in contrast to the northern part, which is characterised by rocky areas and cliffs jutting out over the sea. The Romanian Black Sea coast is characterised by the Danube Delta in the north and a mild coast in the south. In a narrow coastal area, the climatic differences are mainly influenced by the proximity to the sea [46].

According to the Köppen–Geiger climate classification, temperate (C) and continental (D) climate groups prevail throughout the study area [46]. The coastal belt of the study area is characterised by a Mediterranean climate (Csa), with hot, dry summers and cool, wet winters, which are typical for the narrow coastal strip encompassing Croatia, Bosnia and Herzegovina, Montenegro, Albania, and Greece. In this area, on the coast of the Adriatic, Ionian, and Aegean Seas, this Mediterranean climate penetrates inland through natural passages and river valleys (e.g., the Neretva valley from the seacoast to Mostar). In the area of Slovenian and Croatian Istria, in the interior of the Dalmatia region, on the western Black Sea coast, in parts of the interior of Serbia, Bosnia and Herzegovina, Bulgaria, and Romania, there is a moderately warm, humid climate, with hot summers (Cfa) and an oceanic climate, also known as a maritime climate, a subtype of the humid temperate climate (Cfb), which is generally characterised by cool summers and mild winters. In addition to the Mediterranean and oceanic climate, many areas in the interior of the Balkan Peninsula have a humid continental climate (Dfa and Dfb), in which there is usually no dry season and rainfall is evenly distributed throughout the year. The Bulgarian Black Sea coast has a humid subtropical climate (Cfa), with considerable maritime and continental influences [46]. Appendix Table A1 shows the variety of climatic conditions for selected locations on the coast and in the interior of the study area.

### 3.2. Host Plants and Their Importance

The European fruit industry, which generates income totalling over EUR 21 billion per year for more than 1.5 million farmers, is under constant threat from invasive alien pests that enter the EU through trade, the movement of goods, and human mobility [47]. Tropical fruit flies are probably the most important group of invasive pests that are frequently intercepted at European ports of entry (20–30% of all interceptions). The importance of the medfly for trade and implementation of quarantine procedures in the international fresh fruit trade is, therefore, undisputed. The medfly, which has been widespread in Mediterranean countries since the last century, appears to be expanding northwards. It is now frequently detected in Central Europe, in areas where it was previously thought to be too cold for populations to survive [48]. Apart from climatic conditions, the biological potential of the medfly depends largely on the number of available host plants and the ripening time of their fruits [6,22,49,50,51,52,53,54]. As the study area in the Balkan Peninsula covers the relevant Mediterranean part and Central Europe, this area is also reflected in the number of potential host plant species (Figure 2) (Table 1) [22,50,55,56,57,58,59,60].

The availability of host plants and the time of fruit ripening are decisive factors influencing the biological potential of the medfly [6,22,49,50,51,52,53,54,55,56,57,58,59,60]. The diversity of potential host plant species in the research area provides the opportunity for targeted interventions and preventive measures [50,55,57,61,62]. The specific host plants have changed over time, with citrus (Citrus spp.), peach (*Prunus persica*), pear (*Pyrus communis*), fig (*Ficus carica),* persimmon (*Diospyros kaki*), apricot (*Prunus armeniaca*), apple (*Malus domestica*), and quince (*Cydonia oblonga*), playing the most important role [57,58,60,62,63,64,65,66,67]. The methods used to assess susceptibility, infestation, and fruit damage vary with respect to time and the preference of different authors, making it difficult to compare infestation data from different locations. Apricot, which is considered a susceptible early seasonal host of the medfly, ripens from the end of May and in June, after the ripening of loquat (*Eriobotrya japonica*) and before the ripening of peaches and nectarines. In northern Greece, apricots reach maturity in early June and low, but significant, infestations are reported for population growth later in the season on late-maturing cultivars, probably due to a small adult population [3,22,57], similar to infestations reported in the Neretva valley in Croatia [53,54].

Peach and nectarine are highly preferred hosts for the medfly in the Mediterranean region and they are the source of considerable damage [1,2,14,22,23,24,25,26,33,36,60,68,69,70,71]. The infestation of peaches in Croatia and Bosnia and Herzegovina can lead to the complete loss of the crop [53,54,55,72]. Peach is a susceptible host plant in southern and central Greece [3,22,57] and also in Montenegro [56]. Peach is also a common host in Albania, although only limited data are available on fruit infestations in this country [58].

Fig is a very susceptible host to the medfly and significant infestations of the fruits are reported every year in Croatia, Greece, Montenegro, Bosnia and Herzegovina, and Albania [22,54,55,56,57,58,73,74]. Sometimes the enormous intensity of an infestation, measured as the number of larvae per fruit, is observed, as in the Neretva valley in Croatia in 2011, when it reached 65.5 larvae per kg of fruit [54]. In Greece, an infestation level of 4–6 larvae per fruit was observed [57]. Figs were also identified as the first seasonal hosts in Romania and Serbia [68,69].

Persimmons have been identified as hosts for the medfly in Croatia, Slovenia, Montenegro, Albania, Romania, and Serbia [50,51,55,56,57,58,59,63,71]. Persimmons play a crucial role in late-season population growth and overwintering in the southern and northern regions of Greece [3,22].

Citrus species are highly favoured and important hosts for the medfly in Greece, Albania, Montenegro, and Croatia [1,2,14,22,23,24,25,26,33,36,60,68,71]. Several citrus species are mentioned as important hosts: sweet oranges and mandarins are the most important hosts from an economic point of view [3,49,53,54,61,64,66,70,73], while bitter oranges are one of the most important overwintering hosts in central and southern Greece, Montenegro, Croatia, and Albania [3,22,56,58].

Apples have become an economically important host since 2005, with significant infestations reported in Bosnia and Herzegovina [55,65,73]. In northern Greece, high infestation rates have been observed in various apple varieties [3,22].

The infestation of apple fruits has also been observed in Montenegro and Croatia [53,54,56]. In Albania, apples are recognised as a host for the medfly, although detailed infestation data are lacking [58].

Among the economically less important hosts that experience less damage is kiwi (*Actinidia deliciosa*), which has been cultivated in small orchards since 1972 [75]. However, due to its late ripening period, infestation by the medfly is also rare [64,72]. Pomegranates (*Punica granatum*) have suffered from economically significant infestations, but infected fruit are usually only discovered after the fruit has broken open, and quinces can also become infested [76,77]. Pineapple guava (*Acca sellowiana*) and jujube (*Ziziphus jujuba*) are sporadic hosts found in some backyard trees in Mediterranean climates [77,78]. Jujube is the most important confirmed host in Bulgaria and Romania [79].

Less important species of the Rosaceae family, such as plum (*Prunus domestica*) and Japanese plum (*Prunus salicina*), have been confirmed as hosts in the absence of more favourable ones [53,57,77]. Tomato (*Solanum lycopersicum*) and grapevine (*Vitis vinifera*) are rare hosts, with a few records in Bosnia and Herzegovina [55,65,71]. Grapevine can be moderately infested in Greece (NTP unpublished). Kumquat (*Fortunella japonica*), which ripens from January to April, was confirmed as a secondary host in 2015, with a single finding [77,80]. In total, 25 plant species, belonging to 10 plant families and 14 genera, were confirmed as hosts for the medfly in the study area.

See Figure 2 for the seasonal availability of the hosts and Table 1 for the confirmed hosts in the Balkan Peninsula.

**Figure 2 insects-15-00975-f002:**
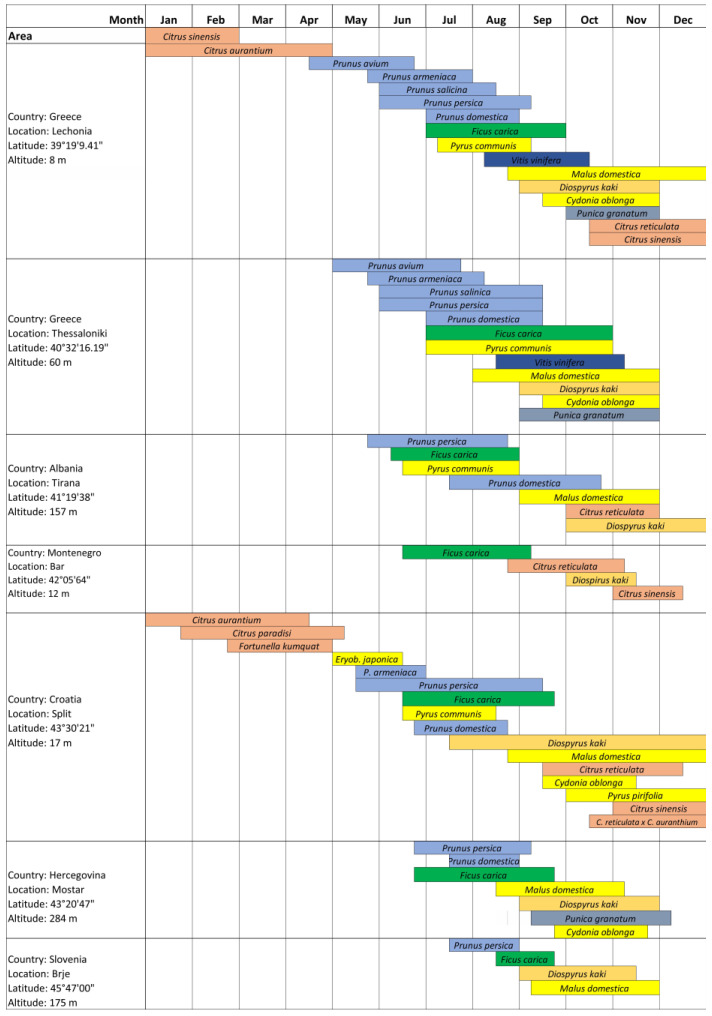
Seasonal medfly host availability (ripening season) in terms of the confirmed fruits in selected characteristic locations along the area of the Balkan Peninsula. Compiled from [3,22,53,56,57,58,59,61,64,67,71,72,73,74,75,77,80].

### 3.3. Historical Overview of Invasion and Current Distribution

#### 3.3.1. The First Occurrence and Introduction Period

The introduction of medfly into a new area occurs mainly through the introduction of infested fruit, which is facilitated by human-mediated transportation and trade [6,8,9]. In contrast to the much earlier historical appearance and spread in the western Mediterranean, the medfly was first recorded in 1914 in Cyprus, an area that is geographically close to the Balkans [81]. The first confirmed report of the pest came from Greece and was dated 1915, specifically in Aigina, and later in the Peloponnese, although its earlier existence cannot be ruled out [82,83]. Since then, medfly detections were reported in coastal areas in Greece and most areas on the mainland, in the central and southern parts (e.g., Athens, Patras, Volos), and on the Greek islands (e.g., Crete, Chios) [84,85,86,87,88,89,90]. In addition, in the last two decades, the medfly has spread throughout the country and reached central Greece, including mountainous coastal areas (e.g., Pelion, Thessaly) and northern and cooler parts, in areas such as Thessaloniki, and in one of the most important deciduous fruit growing areas in Europe, in central Macedonia (Imathia, Pella, Katerini) [91,92,93,94].

There are no data on the route of introduction for the first discovery of the medfly in 1947 in Split, Croatia, followed by several multiple detections along the Adriatic coast in the early 1950s, which indicates a possible source of infection. It is even assumed that the pest was introduced much earlier and adapted to the new environmental conditions before its presence was recorded by both experts and fruit growers [95]. The discovery and outbreak of the pest in northern Croatia, Slovenia, and Montenegro was recorded in the 1950s, and it is assumed that the introduction occurred when a large quantity of infested sweet oranges (Citrus sinensis Osbeck) was imported from Israel and other Middle Eastern countries [96]. The findings from Slovenia (on the border with Italy) suggest that the medfly may have entered this area as a result of natural dispersal from the large fruit warehouses near Trieste (Italy), especially if citrus fruits were stored there for several months [97]. The findings from Croatia for this period describe the high infestation intensity of various hosts, which suggests that the medfly was introduced into this area much earlier. It appears that the pest was only discovered after a naturalisation phase, which may have lasted several years and led to an outbreak involving the infestation of fruit in 1947 [95,96]. The spread and adaptation of the medfly was confirmed in the 1950s within a radius of 10 km from the initial introduction site, when it caused significant damage to peach orchards near Split, Croatia. In 1958–1959, the presence of the pest was confirmed in other coastal areas of Croatia (Crikvenica, Rijeka, Lovran, Medveja, Barbariga, Brijuni Island, the town of Blato on Korčula Island, Dubrovnik, and Opuzen), Slovenia (Koper, Anka-ran, Strunjan, Izola, and Sežana), and Montenegro (Bar) [60,63,66,96,97,98,99,100]. In 1958, an infected peach fruits was reported in the vicinity of Split, Rijeka (Croatia), and Koper (Slovenia), and a significant infestation of persimmon in the areas of Kaštelansko polje, Blato, and Dubrovnik (Croatia), indicating a possible successful establishment of the medfly in the infested areas [60,99]. In 1959, an infestation of peach and a lesser infestation of fig, pear, persimmon, and peach, were recorded at several locations in Croatia (Pod-strana, Splitsko polje, Kaštelansko polje, Stobreč, and Strožanac). These data undoubtedly show that this pest was widespread in the greater Split area in the period 1950–1960 [100]. In the period 1959–1965, the pest was again reported in Greece [83,86,87], in northern and southern areas and in different hosts.

Data from Albania [98,99] clearly show that the medfly was introduced into the commercial area of the harbour in Vlora in 1967 through infested fruit from the mulberry tree (*Morus alba* L.) from southern Italy. Along the eastern Adriatic coast, significant infestations were found in orchards in the 1950s, suggesting that adult flies emerged from the overwintered, infested fruit and infected all the available host plants in the invaded area [66].

The first discovery of the medfly on the Black Sea coast was in 2005 in Varna, Bulgaria [101], followed by detections in Agigea and in the Bucharest region of Romania in 2013 [102]. In Serbia, the medfly was detected in Belgrade and Zemun in 2021 [103]. The detections of the medfly in Romania, Bulgaria, and Serbia, reported since 2000, have been interpreted as a continuous/repeated introduction through infected fruit, according to which the pest infects available hosts during the summer months and can develop one to two generations per year [101,102,103].

#### 3.3.2. Establishment and Spreading Period

Since the mid-1960s, the infestation of various species of host fruit and damage data have confirmed the presence of the medfly in many locations in Slovenia, Croatia, Montenegro, and Greece [66,70,88]. In addition, a significant infestation and damage to mandarins was observed in 1982 near Dubrovnik (island of Lopud and town of Orašac), in Croatia [66], and in the vicinity of Bar and Ulcinj, in Montenegro [69], confirming the continuous spread of medfly. The economic importance of this pest was confirmed in 1990 when fruit damage was observed to a number of economically important host plant species in the coastal region of Croatia [70]. In the late 1990s, the larvae of this pest were detected in export shipments of mandarins from the Croatian Neretva valley on the Slovenian border, leading to national export problems [104,105]. In the 1990s, the medfly was also found to have caused significant damage to fruit in northern Greece (Thessaloniki region, central Macedonia) [22].

#### 3.3.3. Further Spreading Within the Invaded Area

In Greece, the medfly is currently widespread throughout the country, with the exception of the northern, mountainous regions, with strong and long winter periods [68]. The pest was initially reported in three locations, Athens, Aigina, and Leonidion, in the Peloponnese [82,83] and spread through the transportation of infested fruit to central Greece, further to the islands (Crete, Chios, Rhodes) [88,89,90], and finally to the northern parts of the country in central Macedonia [91,92].

In Albania, the pest spread throughout the country in the wider neighbourhood of the cities of Krujë, Tirana, Durres, and Kavaja, and in southern Albania, from Vlora to Saranda [98,99]. In the period 2005–2006, the pest was detected in peach, orange, mandarin, and persimmon fruits [71]. In 2015–2016, surveys showed that the medfly was well established in the areas around the cities of Skhoder, Tirana, Elbasan, Lushnje, Vlore, and Saranda [58,62,106].

In Croatia, in the period from 2000 to 2005, the medfly was present in the entire Dubrovnik–Neretva County, including the upper part of the Neretva valley, up to the border with Bosnia and Herzegovina, in the areas of the island of Korčula and the Pelješac peninsula, in the entire Split–Dalmatia County, including the islands of Hvar, Brač, Vis, Šolta, and Drvenik, and in a part of the coastal area of Šibenik–Knin County, up to the town of Primošten [70,104,105]. In the period from 2006 to 2015, the presence of the pest was again confirmed in the coastal areas of six counties in Croatia, which proves its strong establishment in the area. New findings confirm its further spread along the invaded coastal region in Zadar (Zadar, Sukošan, Bibinje, Turanj, Sveti Filip i Jakov, and Pakoštane) and Šibenik–Knin County (Drage and Brodarica). In addition, medfly has spread into the towns of Polača, Poljica, and Ljubač in Zadar County and in the areas of Vrgoračko polje and Dusina in Split–Dalmatia County. On the Istrian peninsula in Istria County, the pest was detected in almost the entire coastal region (Savudrija, Umag, Poreč, Labinci, Pula, and Vodnjan) and in Čavle, near the town of Rijeka in PrimorskoGoranska County [50,51,52,53,54,74]. In 2015, dozens of medfly adults were caught in an apple orchard in the town of Velika Gorica, near Zagreb [80]. The last survey, conducted in 2020, showed that the medfly had invaded the interior of Split–Dalmatia and Šibenik–Knin counties. Even with a very small number of flies caught, the pest was detected in various places, namely Turjaci, Tugare, and Gata in Split–Dalmatia County and Badanj, Ramljane, and Knin in Šibenik–Knin County [107].

In Bosnia and Herzegovina, a large pest population and a significant infestation of figs were recorded in 2005 [55]. In the period from 2005 to 2010, a large pest population was found in Bosnia and Herzegovina in the area of the Neretva river valley, from the border with Croatia to the city of Mostar and in the areas of Gabela, Višići, Čapljina, Stolac, Buna Mostar, Vrapčići, Željuša, Popovo polje, and the city of Ravno [88]. Recent surveys from 2009–2013 confirm that the medfly is well-established in Bosnia and Herzegovina and is spreading further inland (Potpolje-Čitluk) [72,108].

In Montenegro, the medfly was well-established on the coast and around Lake Skadar, where infestations were observed in mandarin orchards, as well as in the Bay of Kotor, in 2003 and 2004, and in Beri, near Podgorica, in 2010 [56,61,64,69]. In the period 2008–2014, further spreading was confirmed along the entire Montenegrin coast, near the towns of Ulcinj, Bar, Budva, Herceg Novi, and Kumbor [78].

In Slovenia, an infestation of persimmons was detected at the Strunjan site in 2007 [57]. In 2013–2014, the pest was confirmed in figs and persimmons at locations along Piran Bay and Kopar Bay in Izola, Strunjan, and Lucija. In addition, the pest’s reach was extended to new locations inland, where it was discovered in Dekani and Hrvatini [57]. Recent surveys confirm the spread and establishment of the medfly in Cikuti, Brje, Soltan, and Tublje pri Komnu, in the Primorska region, and the pest’s strong preference for persimmon [59].

The fact that the pest has been detected near all the major commercial harbours in Koper (Slovenia), Rijeka, Split and Dubrovnik (Croatia), Bar (Montenegro), and Vlora (Albania), during the 75-year study period, undoubtedly indicates multiple introductions along the Adriatic coast, where it became established and subsequently spread further. The historical distribution of the medfly in the Balkan Peninsula is shown in Figure 3. The currently recorded Medfly population is shown in Figure 4. Appendix Table A2 provides a list of locations where the medfly has been detected, based on the order of historical detections, geographical coordinates, and the countries where it was found.

### 3.4. Pest Phenology

According to the currently available data, the majority of the medfly population in the eastern Adriatic and southern Balkans overwinters mainly in the larval stage in fruits. Recent studies in the Split region (Croatia) confirm the possibility of overwintering of the pests in the adult and pupal stages in protected conditions, such as in the bases of buildings or private houses, but also in field conditions, until the summer months [109]. Adults emerge in early April and the flight of the overwintering generation continues throughout spring and early summer [109]. The first significant feeding events on host fruit are observed in apricots, peaches, and nectarines, from mid-June to the end of August [50,51,52,53,54,73,74]. Although the generations overlap because the adults survive for several months and can infest the fruit of a larger number of hosts, medfly can develop more than two generations in peach fruit [96,100,110,111]. In June and July, rare orchards or individual grapefruit trees (C. paradisi), whose fruits ripen and remain on the trees until late spring, can be more heavily infested than peaches and nectarines [80]. After peaches, nectarines and grapefruits are infested; the next suitable hosts follow, usually figs, plums or summer pears, which ripen from the beginning of August into September. A very large medfly population develops in figs, as they are rarely treated with pesticides [49,50,51,52,53,54]. In mid-September, most of the adults from the fourth generation appear to infest mandarins, from the beginning of ripening until the end of November. During this period, other hosts may also be infested, such as apples, nashi pears, pears, persimmons, clementines, and pineapple guavas [49,50,51,52,53,54,73,74]. The majority of medfly populations in late-season host fruits overwinter as larvae in the fruit and continue their development as pupae in the soil before emerging. The predominant plant species in which medfly overwinter depends on the geographical location. In the Primorska region in Slovenia, in Istria County in Croatia, and in Thessaloniki in Greece, the main overwintering hosts are apples or persimmons [3,22,57,59]; in the coastal areas of the Dalmatia region in Croatia, Albania, southern Greece, and Montenegro, the most common overwintering hosts are citrus species, especially bitter/sour oranges, but also mandarins and sweet oranges [61,69,109,112], while in the area of Bosnia and Herzegovina and northern Greece, the host is mainly apples [3,55,65]. The large number of hosts with different ripening times (Figure 2) is the most important agroecological factor influencing not only the establishment of the pests, but also the population density during the season [22,49,57,58,59,61,62,71,88,89,105].

The data presented in Figure 5 show a significant influence of the latitude on the onset of adult medfly activity (adult captures) and on the duration of flight during the season. In the northern coastal region between 44° and 45°, adult catches in Slovenian and Croatian Istria and in central Macedonia (Greece) start in mid-September and end in mid-November [57,91]. In contrast, adult catches in the coastal region between 42° and 43° in the Dalmatian region of Croatia and Montenegro start in mid-June and last until the end of December and, in some years, rare individuals are caught in January/June of the following year [61,71,88,90,104,105]. The earliest start of the medfly flight is recorded in the area between 39° and 41° latitude in early or mid-May and lasts until the end of the year, as is also the case in Albania and western Greece [58,71,94].

The research results collected over the last ten years, during the period 2014–2022, undoubtedly indicate a very pronounced invasive character of the pest and its ability to adapt to different environmental conditions and spread to new areas. The data presented in Figure 6 show several invasive locations where the medfly has spread from its initial establishment to new neighbouring areas and to higher altitudes [57,59,94,102,106,107,108]. In these cases, it should be emphasised that the combined negative effects and the combination of 43–45° N latitude and the considerable altitude above sea level were not an obstacle for the medfly to invade these less favourable areas. If we also consider that the much higher altitudes where the medfly has spread, from 400 to 600 metres above sea level, are climatically unfavourable, the invasive nature of the pest is undoubtedly confirmed [94,107,108].

In areas where the medfly is established, the pest is usually inactive between January and April and, in recently invaded areas, this absence is prolonged until September, when adults are caught in traps. Even the medfly is unable to overwinter at the egg stage; there is evidence that it can survive low temperatures in fruit hosts as a larva and, to a much lesser extent, in the pupal and adult stages [109]. The onset of adult pest activity in established areas begins between May and August, depending on the location and conditions in the particular year. The peak of the medfly activity is observed in autumn, usually between September and November, and a strong decline is observed between November and December (Figure 5). In recently invaded areas, the flight period is significantly shorter and is at least one and a half months or less during September and October, which overlaps with the peak in the population in the established distribution areas (Figure 6). We, therefore, divided the year into three seasons (January–April, May–August, and September–December), which is important from a phenological perspective for the medfly and its ability to reproduce, while maintaining some common environmental characteristics of the seasons. The seasonal activity of the medfly in the Balkan Peninsula shows that the first four months of the year are much less suitable for medfly activity (Figure 5) [22,89]. During these months, medfly usually overwinter as larvae in host fruits or partly at the pupal stage in the soil, while adults rarely overwinter in urban hotspots [22,89,109]. In the period from May to August, the climatic suitability for the medfly is well-pronounced, while the highest density of medfly is observed in September and October.

### 3.5. Importance of Agroecological Conditions for Future Spreading and Establishment

This review, based on the extensive literature research described above, has provided new insights into the behaviour of medfly under different agroecological conditions. Several different scenarios for the establishment of the medfly and possible indications of its future spread were observed in the study area.

The first scenario is observed in Greece and the eastern part of the Adriatic (Croatia, Montenegro, and Albania). After the first detections (1915 and 1947–1960, respectively), the medfly spread southwards and northwards or along the coastal and island regions, where the conditions were suitable for its development. This trend continued for several decades in the case of Croatia and Greece (Figure 3), where the possibilities for inland spread are given in terms of host availability and climate. However, the medfly continued to spread and only established itself in areas at low altitudes (usually <50 m). Until the year 2000, only very rare detections at higher altitudes were reported. From 2000 onwards, the medfly began to penetrate previously less favourable geographical areas and establish itself at significantly higher altitudes (Figure 7). A similar trend in terms of further spread after initial detection can be observed in the case of Montenegro, where the pest has spread westwards, along the coast, in recent decades (Figure 3). In Albania, the medfly was also discovered on a large scale along the coastal areas, where numerous hosts were available (Figure 3). The occurrence in the Albanian mainland, around Tirana, Kruje, and Elbasan, can certainly not be interpreted as the spread of the medfly, but as an introduction and establishment in areas with altitudes < 200 m (Figure 7). It seems that the medfly follows the pathways and corridors with the least environmental resistance during its progressive invasion, as suggested in regard to California (USA) [113].

The second scenario applies to countries with less favourable conditions for medfly establishment (e.g., Slovenia and Bosnia and Herzegovina). These countries show similar patterns of medfly spread to high altitudes. In Slovenia, the medfly has spread along the coast (Figure 3) and through the Dragonja river valley towards Croatia, where the pest has been documented since 2006–2009. From 2010 to 2020, established populations were found at higher altitudes, up to 150 metres, and in regard to the northernmost confirmed established population (45°8′ latitude). A similar trend can be observed in the Herzegovinian part of Bosnia and Herzegovina. Since the first detections in 2005–2006, the medfly has spread locally along the Neretva valley, over 50 km inland, and has invaded low-lying areas in the valley (Figure 3). A few years after its first discovery, the medfly spread to the higher areas in the Neretva valley and westwards and eastwards out of the Neretva valley, reaching higher altitudes (200–250 m) (Figure 7). It appears that the spread in both Slovenia and Bosnia and Herzegovina follows different patterns than in Greece, Croatia, Montenegro, and Albania.

A third scenario is reported in Bulgaria and Romania and, more recently, in Serbia. In Bulgaria, there was a single detection in Varna in 2005, but no detections on the Black Sea coast. Surprisingly, in the period 2016–2020, the medfly was detected at six locations, along a 300 km long west–east line, with a geographical distribution between 42°1′ and 42°4′ latitude, and at an altitude of 200 to 450 m (Figure 3). In Romania, the medfly was detected several times between 2010 and 2015 and, similar to Bulgaria, it was recorded in three locations at a distance of over 200 km from west to east (44°0′–44°2′ latitude; altitude above <100 m; Figure 3). It seems that this scenario, which includes several relatively recent introduction events, could explain the detection in Bulgaria, Romania, and Serbia in areas that still are at risk in terms of the establishment of the pest (Figure 7).

Although detailed information on the occurrence of the pest and seasonal data for the period after the first detections and establishment is lacking, we observed three trends in the period from 2000 onwards. Firstly, the pest continues to spread along coastal and island regions and occupies a larger area, which is confirmed by the number of new detections. Secondly, it is now occurring in more northerly areas that cross the traditional geographical boundary of the 43rd parallel. The number of detections between the 43rd and 45th parallel has increased over time. Finally, the occurrence of the pest at higher altitudes (200 to 600 m) has been recorded in the last ten years [60,84,88,106,107,108]. The information on confirmed host plants (Table 1) and the seasonal availability of host plants (Figure 2), as well as trapping data as a function of the geographical altitude and elevation above sea level (Figure 5 and Figure 6), serves not only as a basis for recording the history of the medfly invasion and the invasion hypothesis, but also as a basis for predictions on the annual and seasonal distribution of the medfly in terms of suitability. Figure 7 shows the trend in the medfly invasion history along the Balkan Peninsula, as a function of the invasion periods/phases (introduction, establishment, and spread) and horizontal and vertical expansion.

## 4. Discussion

All the available literature on the occurrence and distribution of the medfly in the Balkan Peninsula, both spatially and temporally, is crucial to understand not only the current and historical extent of its occurrence, but also the conditions under which it can survive and the areas that are “vulnerable” to potential invasion and establishment of the medfly and other fruit fly pests of economic importance.

International trade [114] leads to a higher risk of introduction of fruit flies of economic importance and understanding the complex invasion process on a global scale is crucial [115]. The notorious and highly invasive medfly pest has been the subject of extensive research, largely confirming the findings in terms of its historical invasion at a global scale [116,117]. Some of the research in the Mediterranean reports two routes of colonisation. In the period when Europe was closely linked to Africa through trade links [6], the colonisation of Europe took place either through western (via the Iberian Peninsula) or eastern (via Turkey) routes, or a combination of both [116,117,118]. Within Europe, Italy, Croatia, and Greece show more similarities with each other than with Spain, which could be linked to the historical invasion routes of the medfly into Europe and the Balkan Peninsula [119].

Of particular interest are the responses of Tephritid fruit flies to different agroecological conditions. Recent studies have shown that different populations of the medfly [120,121], the fruit fly (*Ceratitis rosa*) [18], and the peach fruit fly (*Bactrocera zonata*) [122] have different tolerances to a variety of temperature conditions, which explains their enormous ability to colonise temperate areas. In European medfly populations, a significant shift in allele frequencies associated with adaptation to cold winter temperatures has been observed, demonstrating the established genetic basis for the high climatic adaptability of medflies to seasonality involving cold (winter) temperatures [119]. Recent experimental studies on the cold tolerance of the medfly, exploring the chill coma recovery time in several European populations, revealed a modest but significant correlation between location and cold tolerance [123]. Recent genetic studies indicate that a genetic predisposition was already present to some degree in the native range, suggesting selection based on existing variation rather than the formation of de novo variants in Europe [119].

The above findings emphasise the potential of invasive fruit flies to rapidly expand their climatic niche in response to climatic conditions during the invasion. For several fruit fly species, there is evidence of adaptation to winter cold stress through changes in allele frequency, which raises the question of whether other invasive Tephritids may be able to adapt to seasonal cold stress in a similar way [119]. However, it is not known how the species respond to the climatic challenges in their new environment. The examples from Greece, Croatia, Slovenia, Bulgaria, and Romania, since 2000, which demonstrate the spread of the medfly to higher altitudes and the presence of established populations, provide some support for the above statements [32,59,94,106]. Records of occurrence with temporal references are important for understanding the drivers of seasonal medfly population dynamics, which can be valuable for targeting eradication and control strategies [41]. To corroborate the abovementioned statements, several detections of medfly have been made in recent decades in the areas to the north and west of the Balkan Peninsula in Austria [31], Germany [32], Romania [32,124], Ukraine [125], Hungary [126], and Russia [127,128]. In addition, transient medfly populations can survive in protected areas that allow the medfly to overwinter in unfavourable temperature conditions [129]. It appears that a series of plastic and adaptive responses and their interactions are responsible for the persistent invasion of the pest in temperate Europe [130,131,132].

The data cited in this present paper, covering the area of the Balkan Peninsula and the occurrence and spread of the medfly in both spatial and temporal terms, are crucial not only for understanding their current distribution, but also for modelling and predicting the further invasion and establishment of the medfly in central and northern Europe and how other Tephritidae fruit flies might follow similar paths. Recently, it has been observed that many tropical fruit flies have expanded their range poleward into cooler Mediterranean and warm temperature regions, likely due in part to climatic changes and altered host availability [133,134]. Several recent outbreaks of fruit flies of economic importance highlight the global invasion issue. One such case is the detection of the Oriental fruit fly (*Bactrocera dorsalis*) in Naples, Italy, in 2018, and in France, in 2019, followed by an outbreak in 2022 in the same area of Naples [135]. At the same time, the peach fruit fly was detected in Israel and North Africa [136]. Due to its potential economic impact, the invasion of this fly into the rest of the EPPO (European and Mediterranean Plant Protection Organisation) area is a growing concern [137]. The ongoing effects of climate change are shifting the potential geographical distribution of the medfly further and further into regions that were previously only suitable for transient populations [137].

The expansion of the medfly’s climatic niche to higher altitudes [32,59,94,106] could have a significant impact on the horticultural industry. This could result in previously unaffected regions now becoming vulnerable to pest infestations. The duration of suitable conditions within a season could increase, resulting in more generations being completed in a season. Increased population densities later in the season could lead to significant infestation rates in later-maturing hosts [137]. The knowledge gained in this review underscores the urgency of understanding the invasion process of fruit flies, which are of economic importance in relation to the possible damage caused as a result of future invasions.

## 5. Conclusions

The results of this review clearly demonstrate that the geographical range of the medfly has expanded from the time of its first introduction to the present day. A significant trend in the spread of the medfly northwards and upwards into areas that are less suitable for their establishment highlight the evolving invasive potential of the pest. Recent findings show that the medfly adapts to different types of shelters or protected niches in order to survive. These results suggest that the species is capable of completing its life cycle in areas far outside its traditional range. Plastic and adaptive responses and their interactions are the main causes of the medfly’s continued invasion into areas with higher altitudes and into temperate Europe. Recent climatic change, the phenotypic plasticity of the medfly, and the species’ ability to recolonise new areas through local, transient populations, are likely to play an important role in the expansion of the species’ existing global distribution. These behavioural characteristics of the medfly provide an insight into the possible behaviour of other invasive species from the Tephritidae family that represent a treat to Europe and the Balkan Peninsula. The fact that several polyphagous species of the Dacinae (Tephritidae) subfamily have a considerable number of identical hosts to the medfly should also be emphasised. The Oriental fruit fly and the peach fruit fly infest different hosts which are important crops in Europe and the Balkan Peninsula. The genetic predisposition of the medfly in its native range to adapt to cold stress in newly invaded areas raises the question of whether exotic fruit flies that are present in the Mediterranean region may behave similarly. The results of this paper is an important contribution to future and effective pest control strategies in times of climate change.

## 6. Future Directions

We recommend that an open-source database of fruit flies of economic importance in regard to their invasion within Europe should be established, as an official source of data by the European Plant Protection Organisation (EPPO), offering data that include pre-border and border inspections and interceptions;We recommend that the National Plant Protection Organisation (NPPO) create an open-source database of post-border detections, with information regarding pest captures, host plants, locations, and pest risk assessments for vulnerable areas;We recommend the development of national action plans based on the International Standard for Phytosanitary Measures to define the management strategy, eradication and suppression actions, the costs of intervention, and ensure the necessary administrative preparation for the use of financial resources provided by the EU in cases of the detection of regulated harmful organisms and the need for eradication or suppression measures.

## Figures and Tables

**Figure 1 insects-15-00975-f001:**
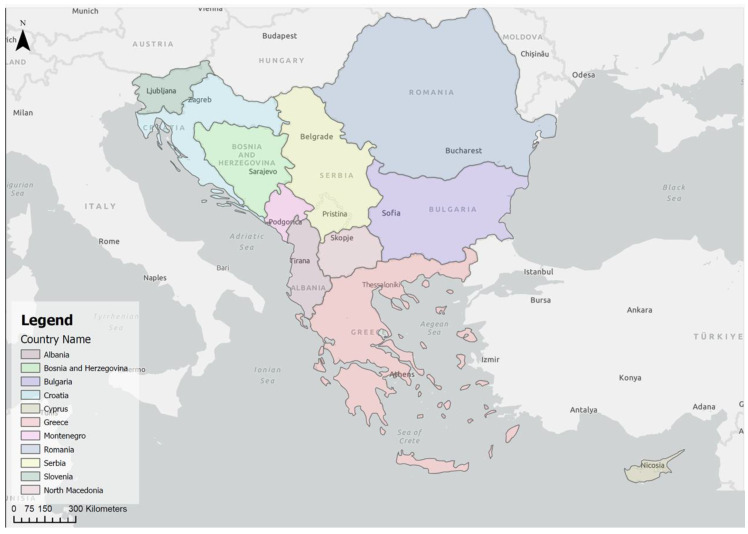
The Balkan Peninsula, which is made up of numerous countries on the European continent (Albania, Bosnia and Herzegovina, Bulgaria, Croatia, Cyprus, Greece, Montenegro, North Macedonia, Romania, Serbia, and Slovenia).

**Figure 3 insects-15-00975-f003:**
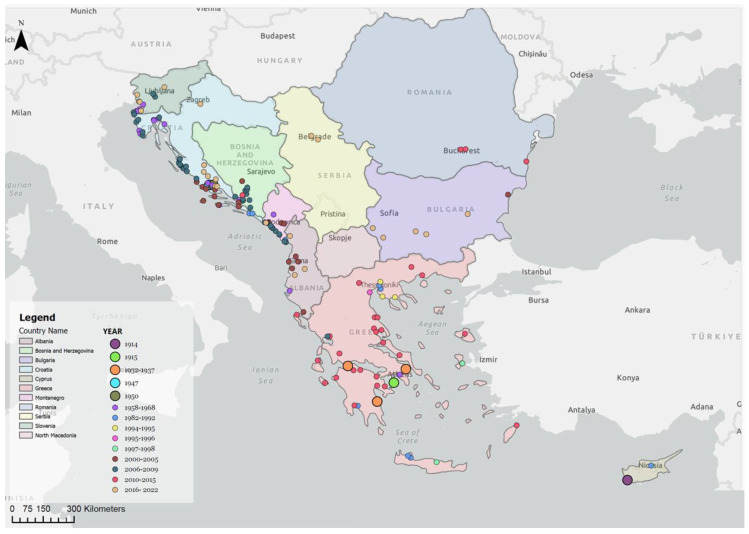
Historical detections of medfly in the Balkan Peninsula.

**Figure 4 insects-15-00975-f004:**
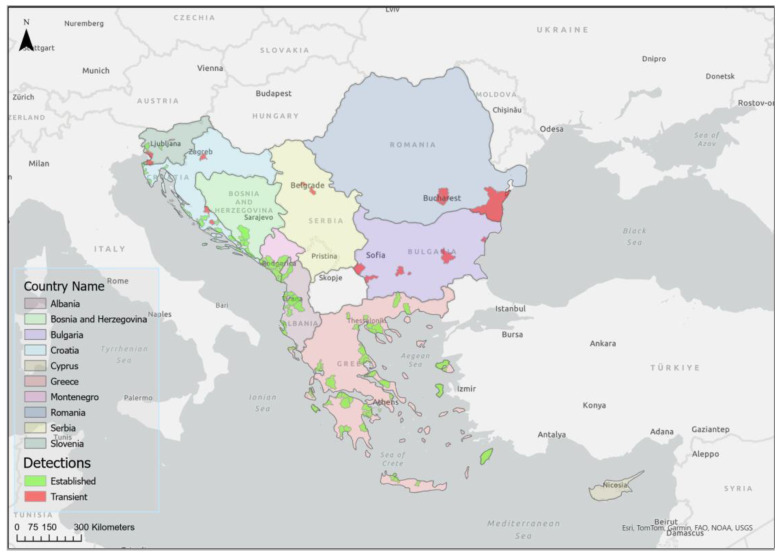
Current distribution of established and transient medfly populations in the Balkan Peninsula, considering the legal status of the pest.

**Figure 5 insects-15-00975-f005:**
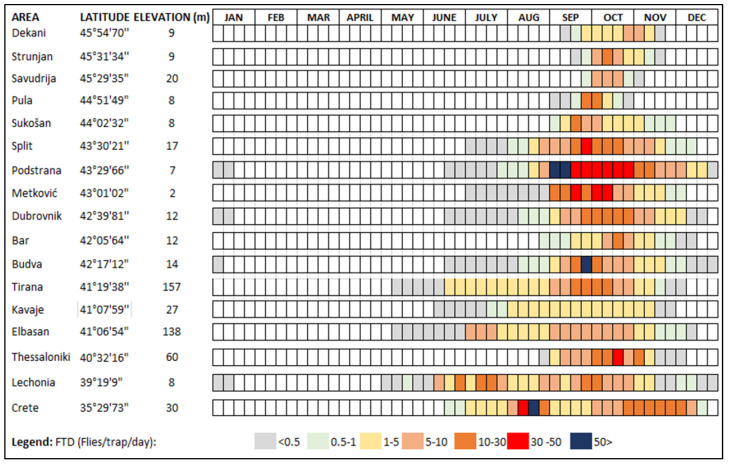
Seasonal medfly flight period in established areas.

**Figure 6 insects-15-00975-f006:**
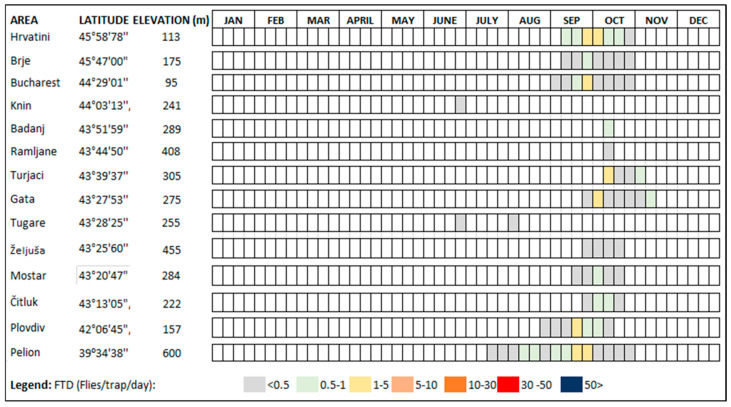
Seasonal medfly flight period in selected inland locations in invaded areas.

**Figure 7 insects-15-00975-f007:**
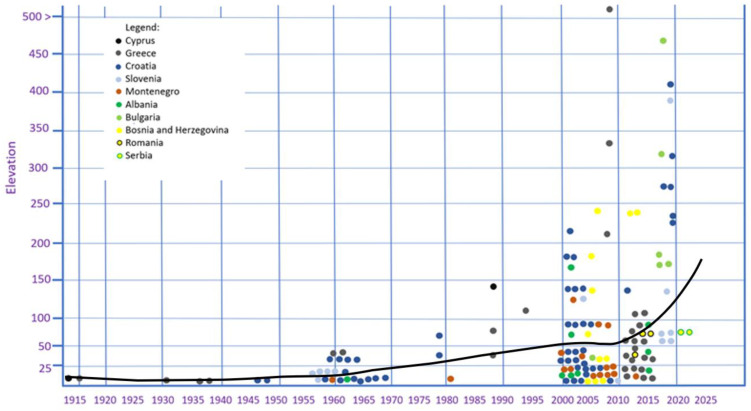
Medfly invasion history and relationship between elevation and year of capture in different countries. (Dots represent detection events over time; the trendline is represented with a loess smoothing line).

**Table 1 insects-15-00975-t001:** List of confirmed hosts of *Ceratitis capitata* in the Balkan peninsula. Legend: SLO—Slovenia, CRO—Croatia, BOH—Bosnia and Herzegovina, MNE—Montenegro, ALB—Albania, GRE—Greece, SRB—Serbia, BUL—Bulgaria, and ROM—Romania (“+” means presence confirmed; “-“means presence is not confirmed).

Plant Family/Host Name	Common Name	SLO	CRO	BOH	MNE	ALB	GRE	SRB	BUL	ROM
Actinidiaceae										
*Actinidia deliciosa*	Kiwi fruit	-	-	+	-	-	+	-	-	-
Ebenaceae										
*Diospyros kaki*	Japanese persimmon	+	+	+	+	+	+	-	-	+
Lythraceae										
*Punica granatum*	Pomegranate	-	-	+	-	-	+	-	-	-
Moraceae										
*Ficus carica* var. *sativa*	Fig	+	+	+	+	+	+	+	-	+
*Ficus carica* var. *caprificus*	Wild fig	+	+	+	+	+	+	+	-	-
Myrtaceae										
*Acca sellowiana*	Pineapple guava	-	+	-	+	-	-	-	-	-
Rhamnaceae										
*Ziziphus jujuba*	Jujube	-	+	+	+	-	-	-	+	+
Rosaceae										
*Cydonia oblonga*	Quince	-	+	+	-	-	+	-	-	-
*Malus domestica*	Apple	+	+	+	+	+	+	-	-	-
*Prunus armeniaca*	Apricot	+	+	+	-	-	+	-	-	-
*Prunus domestica*	Plum	-	+	-	-	+	+	-	-	-
*Prunus persica*	Peach and nectarine	+	+	+	+	+	+	+	-	-
*Prunus salicina*	Japanese plum	-	+	-	-	-	+	-	-	-
*Prunus avium*	Sweet cherry	-	-	-	-	-	+	-	-	-
*Pyrus communis*	Pear	+	+	+	-	+	+	-	-	-
*Pyrus pyrifolia*	Nashi pear	-	+	-	-	-	-	-	-	-
Rutaceae										
*Citrus aurantium*	Sour orange	-	+	-	+	+	+	-	-	-
*Citrus limon*	Lemon	-	-	-	+	-	-	-	-	-
*Citrus medica*	Citron	-	-	-	+	-	-	-	-	-
*Citrus paradisi*	Grapefruit	-	+	-	+	-	-	-	-	-
*Citrus reticulata*	Mandarin	+	+	+	+	+	+	-	-	-
*C. reticulata x C. aurantium*	Clementine	-	+	-	-	-	-	-	-	-
*Citrus sinensis*	Sweet orange	-	+	-	+	+	+	-	-	-
*Fortunella japonica*	Kumquat	-	+	-	-	-	-	-	-	-
Solanaceae										
*Solanum lycopersicum*	Tomato	-	+	+	-	-	-	-	-	-
Vitaceae										
*Vitis vinifera*	Grape	-	+	+	-	-	+	-	-	-

## Data Availability

Data are contained within the article and Appendix A. The original contributions presented in this study are included in the article. Further inquiries can be directed to the corresponding author.

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
