# Peer review of "Invasion History and Dispersion Dynamics of the Mediterranean Fruit Fly in the Balkan Peninsula"

_insects, 2024, doi:10.3390/insects15120975_

Round 1

Reviewer 1 Report

Comments and Suggestions for Authors

The manuscript provides very valuable information on the historical establishment and spread of the Mediterranean fruit fly in the Baltic region. Most of the figures and data is highly informative, and summarizes the Medfly trends in the region, and historical patterns of invasion and establishment. The manuscript is valuable, but it is my impression that it requires a major revision and organization. The manuscript resembles more an assay, than an experimental study, thus, sections need to be modified and adapted to the format of an assay manuscript. The manuscript is also extremely repetitive, and many sections can be joined and importantly condensed. Following some major comments summarizing my suggestions:

1)      Although English in general is comprehensible, it requires proofreading. English in the Abstract and simple summary require major corrections, while other sections require to look for typos and syntactic errors.

2)      The Introduction is in general good, and provides readers with the aims and importance of the assay.

3)      Line 106: Suggest to change “Literature Review” to “Source of Data and organization of assay”

4)      In Line 111-114, I suggest to change the first sentence into “The assay is organized into the following topics: (1) Geographic and climatic characterization of the Balkans Region

5)      I suggest to remove lines 115-144 and integrate them into the actual sections in part 3.

6)      Remove Results (line 145). I don’t see this as a Result section properly. The authors can probably change Result for other term, which may be more appropriate, such as “Analysis of Data”.

7)      Line 146, instead of Study area, will suggest using “Geographic and climatic characterization of the Balkan Region”. Maybe, there are many details on the geography, especially for readers from other regions, which can be condensed.

8)      Section 3.2 has many affinities with section 4.1, and becomes redundant. I suggest to fuse the two and remove section 3.2 altogether. I suggest to keep section 3 only for the geographic and climatic characterization (3.1), and leave the discussion on hosts (3.3) with Fig. 2 and Table 1, which are highly informative. Condense the whole section and remove many details from the description.

9)      To make Fig. 2 more interesting and useful, probably colors can be used. Suggest that bars of the same species of host will be colored with the same color.

10)   Keep section 4.1 (and add discussion from 3.2). I liked section 4 and 5.

11)   Line 401: explain “by accident”

12)   Figure 7 is a graph showing the relationship between time and elevation above sea level. I will state this relationship in the caption. Also in the caption, the use of hypothesis seems to me out of place. The figure is showing the relation between elevation and years of captures in different countries, not hypothesis. I will change the whole caption, and be more explicit.

13)   The Discussion and Conclusion sections are a repetition of all the previous sections. The two sections can be completely removed. I suggest to concentrate on the usefulness of the data presented in the assay to model and forecast medfly invasion to Central and Northern Europe, and how other fruit flies may follow similar paths. That is, the discussion and conclusions should not be a summary of the previous sections, but provide some insights that can be useful to predict and/or extrapolate to other regions.

Comments on the Quality of English Language

Requires proofreading

Author Response

1)      Although English in general is comprehensible, it requires proofreading. English in the Abstract and simple summary require major corrections, while other sections require to look for typos and syntactic errors.

DONE: Both, Abstract, Simple summary and the text were proofreaded

2)      The Introduction is in general good, and provides readers with the aims and importance of the assay.

WELL RECIEVED

3)      Line 106: Suggest to change “Literature Review” to “Source of Data and organization of assay”

DONE

4)      In Line 111-114, I suggest to change the first sentence into “The assay is organized into the following topics: (1) Geographic and climatic characterization of the Balkans Region

DONE

5)      I suggest to remove lines 115-144 and integrate them into the actual sections in part 3.

DONE

6)      Remove Results (line 145). I don’t see this as a Result section properly. The authors can probably change Result for other term, which may be more appropriate, such as “Analysis of Data”.

DONE

7)      Line 146, instead of Study area, will suggest using “Geographic and climatic characterization of the Balkan Region”.

DONE

Maybe, there are many details on the geography, especially for readers from other regions, which can be condensed.

AUTHORS WANT TO KEEP IT THIS WAY; OTHER REVIEWER ON ANOTHER HAND WANT TO EXTEND IT, WE WOULD LIKE TO KEEP IT ON SUTCH EXISTING LEVEL.

8)      Section 3.2 has many affinities with section 4.1, and becomes redundant. I suggest to fuse the two and remove section 3.2 altogether. I suggest to keep section 3 only for the geographic and climatic characterization (3.1), and leave the discussion on hosts (3.3) with Fig. 2 and Table 1, which are highly informative. Condense the whole section and remove many details from the description.

DONE, For the 3., we change title to Characterisation of the area: 3.1. Geographic and climatic characterisation and 3.2. Host plants and their status. Furtermore, we merge „old 3.2.“ with 4.1.

Host status is condensed becouse of the lack of the dana from litareture abouth methodology of infestation assesment

9)      To make Fig. 2 more interesting and useful, probably colors can be used. Suggest that bars of the same species of host will be colored with the same color.

DONE: We made visualisation for Citrus, Pome fruits, stone fruits and other single hosts

10)   Keep section 4.1 (and add discussion from 3.2). I liked section 4 and 5.

DONE

11)   Line 401: explain “by accident”

Removed

12)   Figure 7 is a graph showing the relationship between time and elevation above sea level. I will state this relationship in the caption. Also in the caption, the use of hypothesis seems to me out of place. The figure is showing the relation between elevation and years of captures in different countries, not hypothesis. I will change the whole caption, and be more explicit.

DONE: Beside change of the title to „relation“ that is more appropriate, the title of the chapter is changed to new one: Importance of agroecological conditions for future spreading and establishment.

13)   The Discussion and Conclusion sections are a repetition of all the previous sections. The two sections can be completely removed. I suggest to concentrate on the usefulness of the data presented in the assay to model and forecast medfly invasion to Central and Northern Europe, and how other fruit flies may follow similar paths. That is, the discussion and conclusions should not be a summary of the previous sections, but provide some insights that can be useful to predict and/or extrapolate to other regions.

DONE: We made new Discussion chapter and new conclusion to avoid repetition, thanks for this suggestion

Reviewer 2 Report

Comments and Suggestions for Authors

The manuscript addresses a literature review on important aspects of the medfly invasion in the Balkan Peninsula. It allows for the exploration of articles in several languages, in addition to English, which expands local knowledge to a global level.

I suggest using keywords other than the title.

Section 2. Literature review and research activities should detail the databases used (Scopus? Web of Science?), the keywords used, and the exclusion and inclusion criteria.

The 'Study area' section should be renamed as 'Geographic Scope of the Review' to better reflect its content. This change will help readers quickly understand the section's focus. I suggest including in this section which fruits are produced in each country, the area cultivated with fruit trees in hectares, and the value of production and export. The geographical description is very long, I suggest that these descriptions be summarized, as it makes reading tiring for readers and this information can easily be found by readers in other publications.

I suggest including an item on pest management in the region, specifically the management methods used in each country. This addition will provide a comprehensive overview of the medfly invasion and its management, enhancing the manuscript's relevance to the field of entomology and agriculture.

It would be beneficial for the author to provide a more detailed characterization of the research methodologies used in addressing the infestation of medfly in fruits. This would ensure a thorough evaluation of the work, which likely employs a variety of methodologies, and demonstrate the seriousness with which the work is being reviewed.

The same should be done when referring to population dynamics; in the articles consulted, the authors monitored the species using food attractants or trimedlure.

I suggest changing the name of the final section from 'Conclusion' to 'Future Directions '. This change will better reflect the section's content, which should focus on the current situation, the impact of climate change, and how countries can deal with the problem, highlighting management possibilities, mainly with low environmental impact strategies, such as biological control.

Author Response

Beside answers that are given in the track-changes in the MS where Rev. 2 made comments, bellow are answers to the Rev 2 comments:

The manuscript addresses a literature review on important aspects of the medfly invasion in the Balkan Peninsula. It allows for the exploration of articles in several languages, in addition to English, which expands local knowledge to a global level.

  1. I suggest using keywords other than the title.

DONE

  1. Section 2.Literature review and research activities should detail the databases used (Scopus? Web of Science?), the keywords used, and the exclusion and inclusion criteria.

DONE

  1. The 'Study area' section should be renamed as 'Geographic Scope of the Review' to better reflect its content. This change will help readers quickly understand the section's focus.

DONE

  1. I suggest including in this section which fruits are produced in each country, the area cultivated with fruit trees in hectares, and the value of production and export.

Authors give opinion: Authors opinion is that entering into economic assessment of the production type size, export would be broad for the new paper and dealing also with other issues like border rejections etc. So authors would like to keep content fitting to the title and in this case, still give elementary information.

  1. The geographical description is very long, I suggest that these descriptions be summarized, as it makes reading tiring for readers and this information can easily be found by readers in other publications.

ANSWER: Still, authors opinion is that in one page we give elementary information about the area for the reader and would like to keep it this way.

  1. I suggest including an item on pest management in the region, specifically the management methods used in each country. This addition will provide a comprehensive overview of the medfly invasion and its management, enhancing the manuscript's relevance to the field of entomology and agriculture.

Answer: Pest management section including prevention, detection, monitoring and suppression would bring data for the second paper and so far authors prefer to fit the title in this MS.  

  1. It would be beneficial for the author to provide a more detailed characterization of the research methodologies used in addressing the infestation of medfly in fruits. This would ensure a thorough evaluation of the work, which likely employs a variety of methodologies, and demonstrate the seriousness with which the work is being reviewed.

Answer: Since the methodology data are insufficiently detailed and incomparable, the chapter is reformulated and reduced.

  1. The same should be done when referring to population dynamics; in the articles consulted, the authors monitored the species using food attractants or trimedlure.

Answer: Authors took the published FTD’s for the countries as literature stated; we are aware that different lures can be used; still the main purpose demonstration FTD values in established and newly invaded areas is presented.

  1. I suggest changing the name of the final section from 'Conclusion' to 'Future Directions '. This change will better reflect the section's content, which should focus on the current situation, the impact of climate change, and how countries can deal with the problem, highlighting management possibilities, mainly with low environmental impact strategies, such as biological control.

DONE

Reviewer 3 Report

Comments and Suggestions for Authors

The review organically and comprehensively reconstructs the invasion history of medfly, a primary pest of orchards, in a wide geographic area of Europe. The article is written in a fairly clear and readable manner, and it outlines a number of consequences for the coming years. The introduction frames the topic well, accompanied by extensive and comprehensive references. I have no major remarks to make, minor remarks are given below.

Abstract:

-line 40: add “been” before “detected”

2. Literature review and research activities: it is not clear if Authors presented only published data or added new data. If so, they should be clearly identified when used in the manuscript.

3.1 Study area, lines 185-200: it would be worth to add info about the temperatures caracterizing the different climates reported here.

3.3 Host plants and their importance: lines 254 and followings: put the scientific names of plants in brackets

4. Historical overview…

Line 333: “are to Balkans” is “area to Balkans”?

Line 356: “M. alba” not necessary to repeat, erase

Line 364: “C. reticulata” not necessary to repeat, erase

Line 400: “M. domestica” not necessary to repeat, erase

Line 424: Figure 2 is maybe Figure 3?

Line 431: Figure 3 is maybe Figure 4?

5. Pest phenology

Line 450: “C. paradise” not reported in the Host plant section. Add there or put in brackets

Line 457: “P. pyrifolia” not necessary to repeat, erase

Author Response

Abstract:

-line 40: add “been” before “detected”: DONE

  1. Literature review and research activities: it is not clear if Authors presented only published data or added new data. If so, they should be clearly identified when used in the manuscript.

Done, reformulated

3.1 Study area, lines 185-200: it would be worth to add info about the temperatures caracterizing the different climates reported here.: Temperature data are presented in the Appendix 1

3.3 Host plants and their importance: lines 254 and followings: put the scientific names of plants in brackets: DONE

  1. Historical overview…

Line 333: “are to Balkans” is “area to Balkans”?DONE

Line 356: “M. alba” not necessary to repeat, erase DONE

Line 364: “C. reticulata” not necessary to repeat, erase DONE

Line 400: “M. domestica” not necessary to repeat, erase DONE

Line 424: Figure 2 is maybe Figure 3? DONE

Line 431: Figure 3 is maybe Figure 4? DONE

  1. Pest phenology

Line 450: “C. paradise” not reported in the Host plant section. Add there or put in brackets, Citrus paradisi is listed in Table 1 as a host, and in brackest for the first mention

Line 457: “P. pyrifolia” not necessary to repeat, erase, DONE

Reviewer 4 Report

Comments and Suggestions for Authors

Dear AA., my own compliments for a well written and interesting manuscript.

I would like to congratulate you on this review work that was done with scientific rigor and a complete bibliography. The manuscript flows well in reading and is interesting even for the reader who is not really familiar with the issue of the fruit fly C. capitata and its fearsome danger to agricultural production. This work also opens up the possibility of carrying out simulations for the near future in view of the continuous climate change and therefore it is a starting point for future investigations. I have found only 2 errors:

1) line 212: mountainous insted of mountainoius

2) line 372: put in number the reference.

Author Response

Dear Reviwer 4,

Thanks for the review of our MS.

Al Your suggestions are welcome and changes are done in MS